# Argon Plasma Surface Modified Porcine Bone Substitute Improved Osteoblast-Like Cell Behavior

**Cheuk Sing Choy [1,2], Eisner Salamanca [3], Pei Ying Lin [3], Haw-Ming Huang [1,3,4], Nai-Chia Teng [1,5], Yu-Hwa Pan [3,6,7,8,\*] and Wei-Jen Chang [3,9,\*]**

1 Department of Community Medicine, En Chu Kong Hospital, New Taipei City 237, Taiwan; prof.choy@gmail.com (C.S.C.); hhm@tmu.edu.tw (H.-M.H.); dianaten@tmu.edu.tw (N.-C.T.)
2 Yuanpei University of Medical technology, Hsin Chu, Taipei 300, Taiwan
3 School of Dentistry, College of Oral Medicine, Taipei Medical University, Taipei 110, Taiwan; eisnergab@hotmail.com (E.S.); payinglin53@gmail.com (P.Y.L.)
4 Graduate Institute of Biomedical Materials & Tissue Engineering, College of Oral Medicine, Taipei Medical University, Taipei 110, Taiwan
5 Dental Department, Taipei Medical University Hospital, Taipei 110, Taiwan
6 Department of General Dentistry, Chang Gung Memorial Hospital, Taipei 105, Taiwan
7 Graduate Institute of Dental & Craniofacial Science, Chang Gung University, Taoyuan 333, Taiwan
8 School of Dentistry, College of Medicine, China Medical University, Taichung 404, Taiwan
9 Dental Department, Taipei Medical University, Shuang-Ho hospital, Taipei 235, Taiwan
* Correspondence: shalom.dc@msa.hinet.net (Y.-H.P.); cweijen1@tmu.edu.tw (W.-J.C.); Tel.: +886-2-2736-1661 (ext. 5148); Fax: +886-2-2736-2295 (Y.-H.P. & W.-J.C.)

**Abstract:** Low-temperature plasma-treated porcine grafts (PGPT) may be an effective means for treating demanding osseous defects and enhance our understanding of plasma-tissue engineering. We chemically characterized porcine grafts under low-temperature Argon plasma treatment (CAP) and evaluated their biocompatibility in-vitro. Our results showed that PGPT did not differ in roughness, dominant crystalline phases, absorption peaks corresponding to phosphate band peaks, or micro-meso pore size, compared to non-treated porcine grafts. The PGPT Ca/P ratio was 2.16; whereas the porcine control ratio was 2.04 ($p < 0.05$). PGPT's [C $1s$], [P $2p$] and [Ca $2p$] values were 24.3%, 5.6% and 11.0%, respectively, indicating that PGPT was an apatite without another crystalline phase. Cell viability and alkaline phosphatase assays revealed enhanced proliferation and osteoblastic differentiation for the cells cultivated in the PGPT media after 5 days ($p < 0.05$). The cells cultured in PGPT medium had higher bone sialoprotein and osteocalcin relative mRNA expression compared to cells cultured in non-treated porcine grafts ($p < 0.05$). CAP treatment of porcine particles did not modify the biomaterial's surface and improved the proliferation and differentiation of osteoblast-like cells.

**Keywords:** argon plasma treatment; porcine bone graft; biological apatite; chemical properties; in-vitro behavior

## 1. Introduction

Reconstruction of osseous defects, as performed in the fields of Periodontics, maxillofacial surgery, and orthopedics, can often be achieved by using autogenous bone. Autotransplantation of human bone has been documented since the early 19th century, and autologous bone grafts are considered the gold standard for bone regeneration due to their osteogenic, osteoinductive, osteoconductive, and osseointegrative characteristics [1,2]. However, it is challenging to use only autogenous bone when these defects are large, complex, or require adequate bone volume at the desired location. Availability

(which in some cases is shallow) and autograft acquisitions carry a considerable patient burden, including additional surgical incisions, increased postoperative morbidity, weakened donor bone sites, and potential complications [3]. These adverse effects necessitate the use of alternative materials that mimic the physicochemical and biological performance of natural bone-derived apatite [4–7]. Examples of such alternatives include allografts, alloplastic materials, or xenografts [8]. Unfortunately, fresh bone allografts carry a risk of rejection by the host immune system. Alloplastic materials are either more complex or costly than biological apatite that is directly prepared from animal hard tissues (e.g., bovine, porcine, and cuttlefish bone) [9].

The first reported transfer of fresh bone during cross-species transplantation was described by van Meekeren in 1668 [10]. Nowadays, porcine bone is considered to most closely resemble human bone in terms of their macrostructure and microstructure [11], chemical composition, and remodeling rate [12]. Porcine grafts are also abundant, making it an excellent candidate bone graft material for reconstructing osseous defects [13]. To improve the performance of the porcine bone graft material, various attempts have been made in order to modify the graft material. For example, the addition of fluoride [9] may enable both the fluorapatite formation as well as the antibacterial function of the materials [14].

Plasma discharge applications are based on different geometries and use various electrode materials [15]. In plasma discharge, a source gas is dissociated and ionized, and various particles (e.g., electrons, ions, atoms, radicals, and ultraviolet (UV)) are inactivated following contact with a biological system. Common gases such as Ar, $N_2$, and $O_2$ are used for gas plasma sterilization, thus assuring that no toxic chemicals remain on the objects after treatment [16]. A much-debated topic in the field of tissue engineering is how to define the plasma treatment "dose". Plasma appears to operate over multiple pathways, and the type of device and target tissue will affect the treatment outcomes [17].

Tissue engineering in reconstructive surgery demands the 3-dimensional (3D) growth of osteoblasts and chondroblasts onto suitable carriers [2,18]. This necessitates the fabrication of multifunctional scaffolds that meet structural, mechanical, and nutritional requirements. These scaffolds are used to direct 3D tissue ingrowth for repairing large, complex, and multi-tissue defects [19,20]. A 2017 study found a lack of common understanding regarding the interaction between plasma and living cells, tissues, and organisms. This knowledge gap represents a significant obstacle to developing large-scale clinical trials of plasma-related medical devices and procedures [17]. The use of plasma-treated porcine grafts (PGPT) could enable better 3D reconstruction of challenging osseous defects. PGPT may enhance our understanding of plasma-tissue engineering further. Consequently, we sought to physically and chemically characterize porcine grafts under low-temperature Argon plasma treatment, described previously as cold atmospheric plasma (CAP) [21], and evaluate their biocompatibility in-vitro.

## 2. Materials and Methods

### 2.1. Sample Preparation

Porcine grafts were described previously in another study performed by the same laboratory. Briefly, the porcine graft particles ranged from 500–1000 μm in size and consisted of cortical porcine bone, originating from special pathogen-free (SPF) porcine long bones (Agriculture Technology Research Institute, Hsinchu, Taiwan). First, the bones were immersed in 0.5 N hydrochloric acid (HCl) for 60 min to remove the organic matrix. Subsequently, the porcine bones were heated with a ramp rate of 5 °C/30 s, and were settled at 800 °C for 2 h. Afterward, for another 2 h, with a different ramp rate of 5 °C/min, the grafts were settled at 1000 °C to remove residual soft tissues and proteins. Next, the pieces were cooled to room temperature, milled to 500–1000 μm particle size, and sterilized using γ-rays [11]. Forty samples, measuring 200 mg each, were used for the control porcine grafts (porcine control) and PGPT.

## 2.2. Low-Temperature Argon Plasma Treatment

The Plasma Jet (PJ; AST Products, Inc., North Billerica, MA, USA) was used to subject porcine particles to low-temperature Argon plasma treatment. The APT was carried out at a power level of 80 W, with a frequency of 13.56 MHz, under a 100 m Torr of pressure over 15 min with Argon plasma placed 10 mm from the porcine graft particles.

## 2.3. Surface Topography Evaluations

Scanning electron microscopy (SEM) images (SU3500, Hitachi, Ltd., Kyoto, Japan) were used to evaluate and compare the surface morphologies of PGPT and the porcine control. Sample particles were prepared for imaging using 25 nm-thick layers of Au/Pd sputter surface coating with a sputtering apparatus (IB-2; Hitachi, Ltd., Tokyo, Japan). High-resolution images were taken at a low pressure and an accelerating voltage of 15.0 kV. The electron beam was focused to a fine point (primary electron) and magnified to 500× and 1500× by secondary electrons.

## 2.4. Energy Dispersive Spectrometry

The samples' surface topography probe measurements were taken with an electron beam covering a 70 μm spectrum, operating at 15 kV. An Energy Dispersive X-Ray Micro Analyzer machine was used for the evaluation (EX-250, HORIBA, Kyoto, Japan). Energy dispersive spectrometry (EDS) was useful for analyzing the percentage weight for each element of the PGPT and porcine control particle grafts ($n = 6$).

## 2.5. X-Ray Photoelectron Spectroscopy

Elemental and chemical surface analyses were performed on PGPT and porcine controls using X-ray Photoelectron Spectroscopy (XPS) measurements, following the protocol by Silversmit et al. The measurements were recorded with a Perkin–Elmer Phi ESCA 5500 system equipped with a monochromated 450 W Al Kα source (Quantera II, ULVAC-PHI Inc, Kanagawa, Japan). The base pressure of the ESCA system was less than $1 \times 10^{-7}$ Pa. All experiments were recorded with a 220 W source power and an angular acceptance of $\pm 7°$. The analyzer axis was oriented at an angle of 45° with the specimen surface. Wide-scan spectra were measured over a binding energy range of 0–1400 eV and a pass energy of 187.85 eV. We recorded the C 1*s*, O 1*s*, Ca 2*p*, and P 2*p* core levels [22].

## 2.6. X-Ray Diffraction Analysis

Powder X-ray diffraction (XRD) was used to analyze the crystalline structures and chemical compositions of PGPT and porcine controls. Diffraction patterns were collected on a PANalytical X'Pert3 Pro system (X'Pert3 Powder, PANalytical Co. Ltd., Almelo, The Netherlands) operated at 60 kV and a 45 mA current with Mo Kα (0.71073 Å) source. Samples were scanned over a range of $10° \leq 2\theta \leq 70°$.

## 2.7. Fourier Transform Infrared Spectroscopy (FTIR) Characterization

Perkin–Elmer Spectrum One Fourier Transform Infrared spectroscopy (Perkin–Elmer Corp., Waltham, MA, USA) was used to collect all spectra. Dry PGPT and porcine control particles were equilibrated at 50% relative humidity, at room temperature, and clamped directly onto the crystal for analysis [23]. The nominal resolution of 4.00 and a number of sample scans equal to 1000 was collected in a range of 450–4000 cm$^{-1}$. A computer running Perkin–Elmer 3.02 software was used to record data.

## 2.8. Cell Culture and Seeding

MG-63 osteoblast-like cells were purchased from the Bioresource Collection and Research Center (BCRC, Hsinchu, Taiwan). The cells were expanded in Dulbecco's modified Eagle's medium (DMEM; HyClone, Logan, UT, USA) supplemented with ʟ-glutamine (4 mmol/L), 10% fetal bovine serum,

and 1% penicillin-streptomycin at 37 °C in a humidified atmosphere containing 95% air and 5% $CO_2$. The confluent cells were sub-cultured to the next passage using 0.05% trypsin—EDTA, up to passage 4. Once 90% confluence cell density was reached, the concentration was adjusted to $1 \times 10^4$ cells/mL and the samples were aliquoted into 24-well Petri dishes (Nunclon; Nunc, Roskilde, Denmark). The same day, DMEM medium was mixed with CPG, porcine graft, or HA/β-TCP at a concentration of 1 g/10 mL. Twenty-found hours later, each test well's medium was removed and substituted for the test media, consisting of the previously described DMEM + CPG, porcine graft, or HA/β-TCP. The same DMEM media first described was used for the control wells. The medium in all wells was changed every 3 days.

## 2.9. Cell Cytotoxicity

Cell cytotoxicity was assessed on days 1, 3, and 5 after adding medium and particle bone substitute to the test wells. Tests were performed according to the Cell Proliferation Reagent manufacturer's instructions (WST-1 Kit, Roche Applied Science, Mannheim, Germany). In brief, the cell medium, prepared as described previously, was replaced with 500 µl of fresh medium. Later, the cells were moved to a 96 well microtiter plate ($5 \times 10^4$ cells/well) for a final volume of 100 µL of the culture medium, from which any remaining particle grafts were absent. Afterward, the cells were incubated for 24 h and 10 µL WST-1 reagent was added to each well, after which the cells were again incubated for 4 h in the same standard culture conditions. Next, each plate was settled for 1 min on a shaker to mix its contents. Subsequently, we used a microplate reader at OD = 420–480 nm, a reference wavelength of 650 nm to measure absorbance samples. The percent cytotoxicity was calculated from the following equation: % cytotoxicity = (100 × (control-sample))/control [24]. Each test was repeated 3 times ($n = 6$).

## 2.10. Alkaline Phosphatase Assay

After cell culture and seeding, alkaline phosphatase activity was performed on days 1, 3, and 5. The cells were washed twice with phosphate-buffered saline (PBS). PBS was removed using suction, and 300 µL of Triton X-100 (BioShop, Canada Inc. Burlington, ON, Canada) was added at a concentration of 0.05%. To induce rupture, the cells were subjected to 3 cycles of 5 min at 37 °C and 5 min at −4°C, after which the samples were placed into 96-well plates. Alkaline phosphatase (ALP) activities were determined by following the Thermo Scientific 1-Step p-nitrophenyl phosphate disodium salt (PNPP) manufacturer's instructions. PNPP was supplied pre-mixed with a substrate buffer and ready-to-use at room temperature. The 1-Step PNPP was gently mixed. Next, 100 µL of the mixture was added to each 96-well and mixed thoroughly by gently agitating the plate. The 96-well plates were incubated at room temperature for 30 min. To stop the reaction, 50 µL of 3M NaOH was added and mixed thoroughly by gently agitating the plate. The absorbance of each well was measured at 405 nm ($n = 6$) using a Multiskan™ GO Microplate Spectrophotometer (Thermo Fisher Scientific, Waltham, MA, USA). Enzymatic activity was normalized to total protein concentration using bovine serum albumin (BSA; Roche, Basel, Switzerland). The standard Bradford (Sigma) method was used to do protein measurements. The ALP activity was compared by plotting OD intensity [25].

## 2.11. Real-Time Polymerase Chain Reaction (PCR)

The assay for 5 days was done after the cell culture and seeding protocol. A NanoDrop ND-1000 spectrophotometer (CapitalBio Nano Q, Beijing, China) was used to quantify the total RNA. Later, RNA was processed with the Novel Total RNA Mini Kit, Cat. No. NR-200 (NovelGene, Molecular Biotech, Taipei, Taiwan) according to the manufacturer's instructions. The cells were trypsinized, harvested, and resuspended in 100 µL PBS and subjected to cell lysis by adding 400 µL NR Buffer and 4 µL S-mercaptoethanol to the sample. RNA binding was performed with 400 µL 70% ethanol and centrifuged at 13,000 rpm. Afterward, the sample was washed and eluted with 50 µL RNase-free water.

Real-time polymerase chain reaction (PCR) was used to quantify expression. Gene expression levels were normalized to the expression of the housekeeping gene GAPDH and expressed as fold changes, relative to the expression of the cell culture in DMEM only. The delta-delta calculation method was used to perform quantification. A Primer-BLAST from the United States National Library of Medicine was used to design forward and reverse primers and probes for bone sialoprotein (Bsp) and osteocalcin (OC) genes [26].

*2.12. Statistical Analyses*

All measurements are presented as mean $\pm$ standard deviation, and normality of the results was analyzed using The Jarque–Bera test. Differences between the PGPT and the porcine control groups were identified using the Student *t*-test and considered significant when $p < 0.05$. Microsoft Excel Professional Plus 2016 (Microsoft Software, Redmond, WA, USA) was used to perform all data analyses.

## 3. Results and Discussion

The aim of the present study was to physically and chemically characterize porcine grafts under low-temperature Argon plasma treatment and evaluate their biocompatibility in-vitro. Porcine grafts were used previously in animal studies (e.g., Bone Regeneration for Sheep's Iliac Crestal Defects), where a corticocancellous porcine bone with a 250–1000 microns particulate mix was used as a scaffold to induce bone regeneration. Here, porcine bone was found to be highly biocompatible and capable of inducing faster and greater bone formation. After four months of healing, bone formation was found around the graft particles within the defects [27].

Porcine grafts have also been used successfully in human studies. Successful in-socket preservation, with the combination of a porcine xenograft and collagen membrane, was utilized to maintain successfully the vertical and horizontal dimensions of four-wall bone defects. The authors reported sufficient bone volume for implant placement in all sites prior to implant placement [28]. In another multicenter single-blind randomized control trial, pre-hydrated collagenated cortico-cancellous porcine bone was compared with cortical porcine bone. During the 4-month analysis, both test groups showed reduced bone loss compared to naturally healing sockets. However, the two grafting materials were not able to preserve the alveolar crest, and final results were approximately 30% less than the estimates after healing [29]. Another similar multicenter randomized controlled clinical trial showed that porcine bone (used with guided bone regeneration) was more effective than the control group without grafting after tooth extraction [30].

*3.1. SEM Surface Morphological Observations*

No significant differences were revealed on the surface morphology of the PGPT and non-treated porcine control surfaces. In addition, PGPT showed no difference in roughness, characterized by isotropic and irregular surface patterns, nor a micro-meso pores size, compared to the porcine control. The PGPT particle surface (Figure 1A,B) had the attributes of the non-treated porcine particle (Figure 1C,D), with rough surfaces and microporous structures.

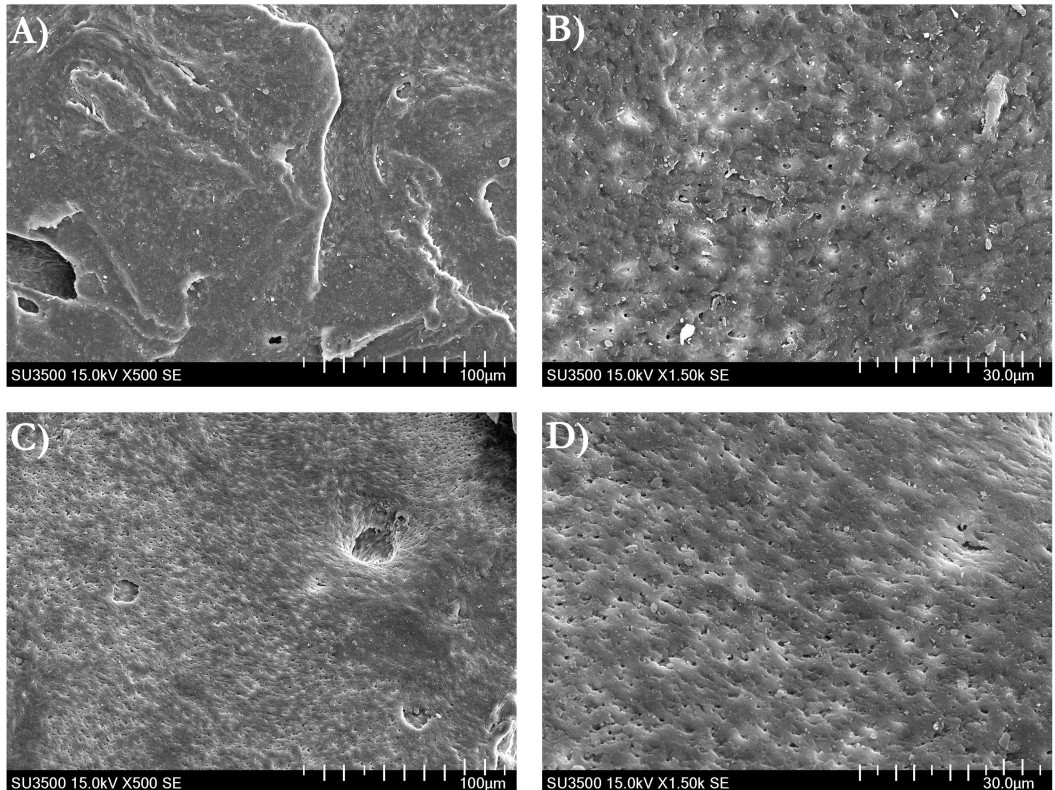

**Figure 1.** The scanning electron microscope images of the representative surface morphologies. (**A**) The rough surface of the plasma-treated porcine graphs (PGPT) is seen at $500\times$ magnification. (**B**) PGPT micro- and mesopores are seen at $1500\times$ magnification. (**C**) Porcine control similar to rough surface PGPT is seen at $500\times$ magnification. (**D**) Porcine control with the same size micro- and mesopores as the PGPT are seen at $1500\times$ magnification.

### 3.2. Element's Weight Percentage on Particles Surfaces

EDS analyses showed that PGPT particle element weight (wt %) contained $37.19 \pm 4.93$ wt % calcium, $34.37 \pm 8.38$ wt % oxygen, $17.19 \pm 1.79$ wt % phosphorus, $4.75 \pm 4.36$ wt % gold, $4.54 \pm 2.62$ wt % carbon, and $0.60 \pm 0.10$ wt % magnesium. By comparison, elemental concentrations in porcine control particles had less calcium with only $30.69 \pm 4.11$ wt % ($p < 0.05$), $15.03 \pm 0.38$ wt % phosphorus ($p < 0.05$), $31.13 \pm 6.98$ wt % oxygen, and $1.13 \pm 0.17$ wt % sodium compared to the plasma-treated porcine particles. The porcine control had a higher level of carbon ($6.02 \pm 6.49$ wt %), magnesium ($1.28 \pm 0.95$ wt %), and gold ($14.72 \pm 2.76$ wt %). The Ca/P ratio in the PGPT particles was 2.16, higher than the hydroxyapatite (HA) value. In contrast, the Ca/P ratio of the porcine control was 2.04, with a statistically significant difference between both Ca/P ratios ($p < 0.05$) (Table 1, Figure 2).

**Table 1.** The results of the elemental analysis by energy dispersive spectrometry.

| Element Weight % | PGPT | Porcine Control |
|:---:|:---:|:---:|
| Ca | $37.19 \pm 4.93$ | $30.69 \pm 4.11$ |
| P | $17.19 \pm 1.79$ | $15.03 \pm 0.38$ |
| O | $34.37 \pm 8.38$ | $31.13 \pm 6.98$ |
| C | $4.54 \pm 2.62$ | $6.02 \pm 6.49$ |
| Mg | $0.60 \pm 0.10$ | $1.28 \pm 0.95$ |
| Au | $4.75 \pm 4.36$ | $14.72 \pm 2.76$ |
| Na | $1.35 \pm 0.53$ | $1.13 \pm 0.17$ |

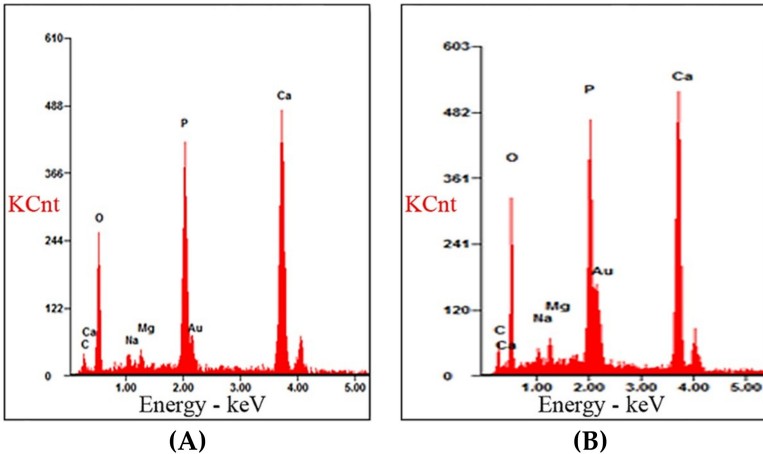

**Figure 2.** The energy dispersive spectra. (**A**) PGPT. (**B**) Porcine control.

### 3.3. XPS Peak Fitting Results

Surface chemistry and atomic concentrations (as percentages), using XPS analysis, showed that the C 1*s* values were 24.3% for the PGPT and 28.5% for the porcine control, indicating a decrease in the number of carbonate groups built into the HAp structure. Moreover, the O 1*s* values were 51.1% (PGPT) and 47.9% (control). The P 2*p* values were 5.6% (PGPT) and 7.1% (control). The mean Ca 2*p* values were 11.0% (PGPT) and 11.11% (control). The Mg 1*s* levels were 6.1% (PGPT) and 2.3% (control). Na 1*s* was lower (1.4%) for PGPT compared with the porcine control (3.2%) (Figure 3).

XPS analysis supported the EDS results and indicated that PGPT values were in concordance with Seo et al. who fabricated HA bioceramics from recycled pig bones by heating to 1000 °C. The HA was composed of calcium, phosphate, carbon and magnesium ions in similar percentage concentrations as the present study's results [31]. In the present study, the Ca/P ratio for the PGPT particles was 2.16, higher than the HA value. In contrast, the Ca/P ratio of the porcine control was 2.04, with a statistically significant difference between both Ca/P ratios ($p < 0.05$). (Table 1, Figure 2) Both Ca/P ratios correspond to high crystalline apatite particles [32,33]. The PGPT higher Ca/P ratio over porcine control ($p < 0.05$) can be directly related to the low-temperature Argon plasma treatment. The Ca/P ratio of both porcine grafts in the present study was higher than the one found by other authors. This can be attributed to the lower temperatures used during the porcine graft preparation in the other studies [31,34].

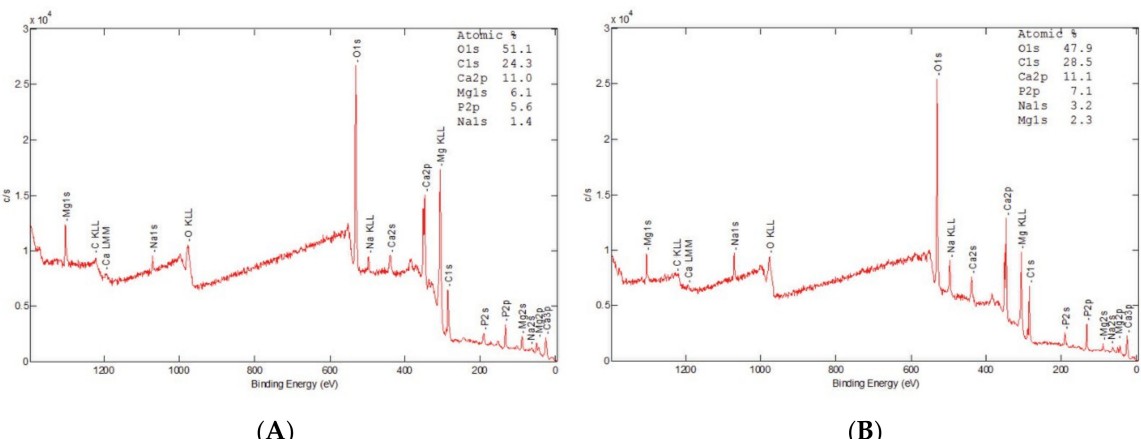

**Figure 3.** The X-ray photoelectron spectroscopy (XPS) spectra used to determine atomic composition (%). (**A**) PGPT. (**B**) Porcine control. Both particle grafts present Ca, P, O, and C.

### 3.4. X-Ray Diffraction Patterns

XRD measurements were compared to diffraction patterns of pure hydroxyapatite, attributing the peaks only to this calcium apatite [31,34]. The porcine control graft remained highly crystalline, as exhibited by the same sharp peak patterns obtained before and after the CAP treatment. The higher-intensity peaks in the patterns correspond to that found in apatites with high crystallinity (Figure 4).

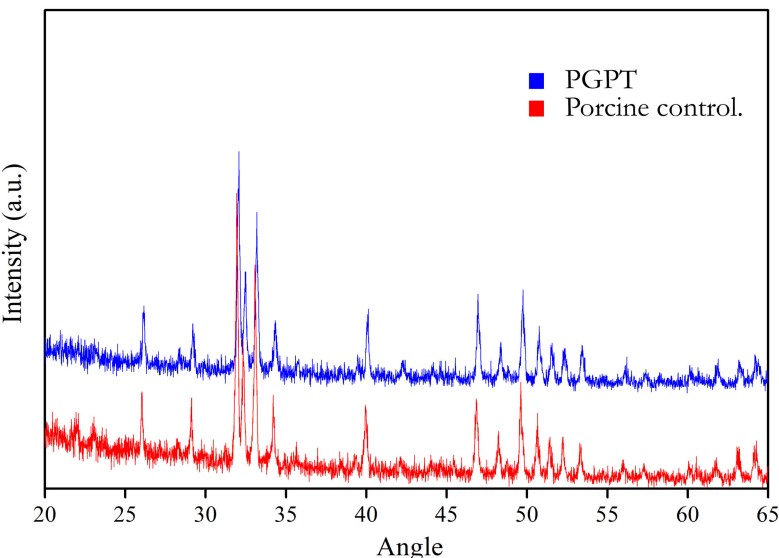

**Figure 4.** The X-ray diffraction patterns. Both porcine particle grafts have the same dominant crystalline phases generating the same intense, sharp peaks.

### 3.5. Fourier Transform Infrared Spectroscopy Profiles

FTIR spectra of sintered PGPT and porcine control particles revealed similar results to those found in the XRD analysis. All samples were characterized as apatite without any other types of crystalline phases. Both samples had similar patterns, with pronounced peaks between 473 and 700 cm$^{-1}$ [35], and absorption peaks ascribed to phosphatase band peaks at 962, 1051, and 1089 cm$^{-1}$ (Figure 5).

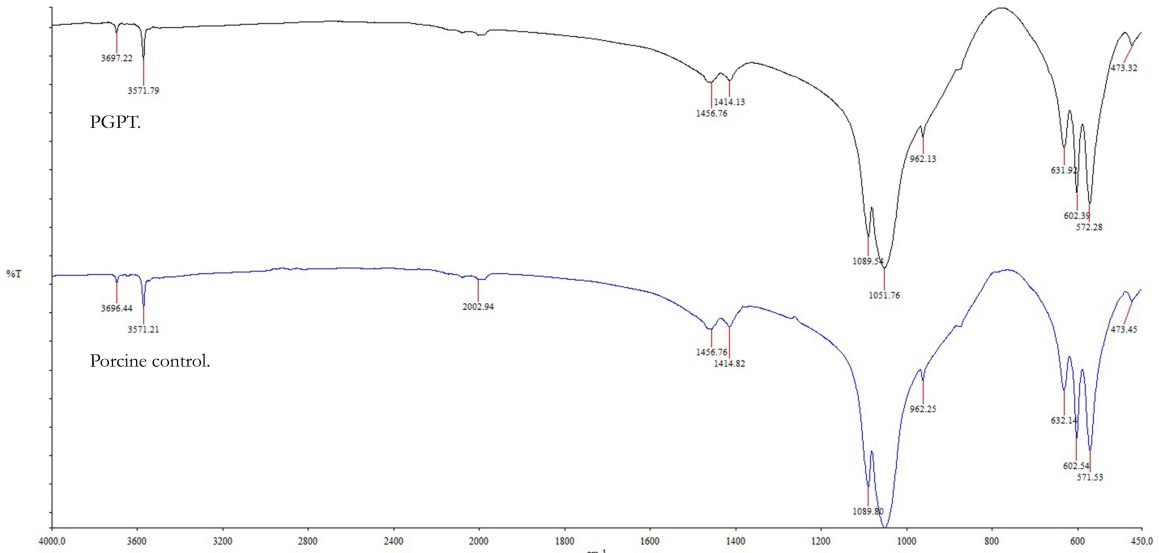

**Figure 5.** The Fourier Transform Infrared Spectroscopy Characterization. Fourier Transform Infrared Spectroscopy (FTIR) spectra of both PGPT and the porcine control with the same absorption peaks corresponding to the phosphate band peaks.

The results of the FTIR and XRD characterization analysis of sintered PGPT and porcine control particles agreed with a study done by Figueiredo et al. which compared the physicochemical properties of bone substitutes used in dentistry vs. calcined human bone. In their study, the diffractograms in the XRD of the porcine graft and natural human bone spectrum presented the more intense characteristic peaks of HA, with coincident peak positions and relative intensities. In their FTIR results, porcine and human graft results showed very similar spectra to the typical bands originated by the HA mineral. More intense phosphate stretching bands were observed around 1010 and 560 cm$^{-1}$ [36]. These were very similar to the FTIR results in the present study, where the same phosphate bands were observed at around 1051 and 572 cm$^{-1}$.

### 3.6. Cell Proliferation Assessment

Cell proliferation OD values on the porcine CAP treated particles were $0.9 \pm 0.05$ (day 1), $1.57 \pm 0.02$ (day 3), and $1.61 \pm 0.08$ (day 5). Conversely, cell proliferation OD values on the non-treated porcine particles were $1.04 \pm 0.04$ (day 1), $1.38 \pm 0.03$ (day 3), and $1.52 \pm 0.01$ (day 5). Cells cultured in the control media had OD values of $1.13 \pm 0.04$ (day 1), $1.46 \pm 0.03$ (day 3), and $1.41 \pm 0.02$ (day 5). MG-63 cells presented with spreading attachments on the PGPT surfaces, improving cell proliferation compared to the non-treated porcine particle surfaces on days 3 and 5 ($p < 0.05$) (Figure 6).

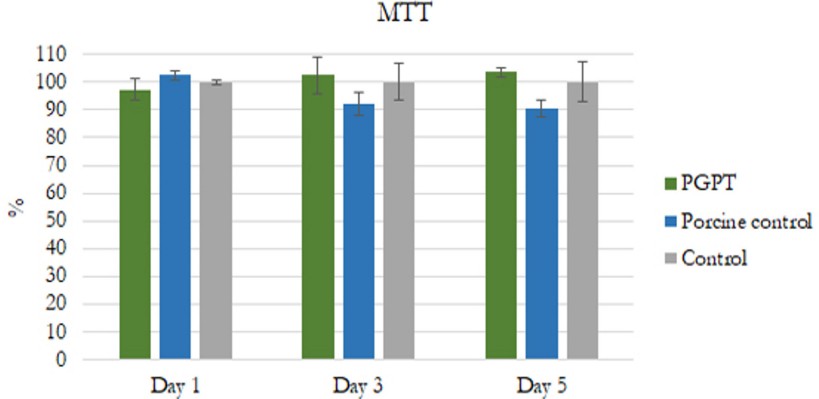

**Figure 6.** The cell proliferation at days 1, 3, and 5 ($p < 0.05$).

### 3.7. Osteoblast Differentiation.

During the time-dependent analysis, the PGPT exhibited higher ALP activity. Significantly greater differences were observed with PGPT than with the non-treated porcine particles and controls on days 3 and 5 ($p < 0.05$) (Figure 7).

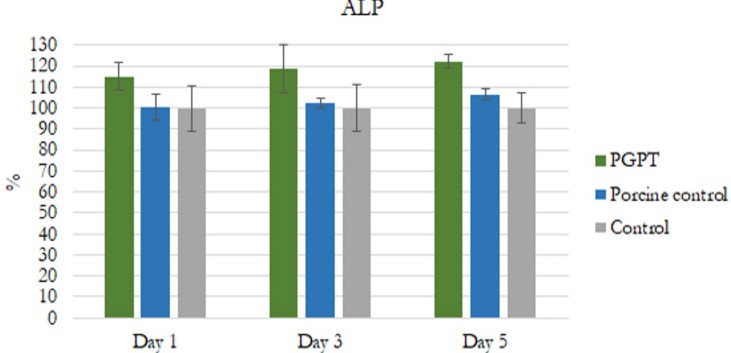

**Figure 7.** The Alkaline Phosphatase (ALP) analysis. Cold Atmospheric Plasma (CAP) porcine treated particles stimulated more alkaline phosphatase production than did porcine control particles in osteoblast-like cells ($p < 0.05$).

After 5 days, the results of real-time PCR revealed that cells cultured with PGPT and porcine control, compared with the control cells, had elevated Bsp and OC genes. In addition, cells cultured with PGPT had a higher relative mRNA expression for both genes than did the cells cultured with porcine control ($p < 0.05$) (Figure 8). To the best of our knowledge, this is the first time that bone sialoprotein and osteocalcin genes have been measured in cells cultured with PGPT. For a better understanding of the porcine bone graft bio-physical surface treatment results within osteoblast-like cells, further analyses of these genes, together with other genes, are necessary.

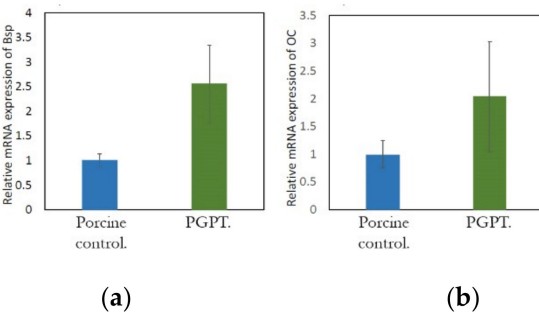

(**a**) (**b**)

**Figure 8.** The real-time PCR. (**A**) Bone sialoprotein mRNA expression of cells cultured at 5 days. (**B**) Osteocalcin of cells cultured at 5 days ($p < 0.05$).

Plasma has gained great scientific and industrial importance. A great advantage of using low-temperature plasmas under atmospheric pressure is our ability to chemically "design" the plasma. For example, plasma composition can be varied depending on the desired effect. In addition, plasma acts rapidly, effectively, and penetrates the smallest openings and hollow spaces [37]. Surface modification of polymers with low pressure is often used to improve coating adhesion, wettability, printability, biocompatibility, and other surface-related properties of polymers or other materials [38,39]. Plasma treatment has also been used for titanium surface modification [40,41]. Sarma et al. examined the biomimetic growth of HA nanocrystals on Ti and sputtered $TiO_2$ substrates [42]. Yi et al. prepared akermanite ($Ca_2MgSi_2O_7$) bioactive coatings using a plasma-spraying technique, with bioactive ceramic coatings on titanium (Ti) alloys [43]. The key question is how to best design and operate the plasma source for optimal biological applications, and this question remains a challenge in the field. This challenge has two parts: first, the physics and chemistry of different plasma devices are far from being fully understood. Second, the mechanisms by which plasma alters biological cells, tissues, and organisms are not well-established [17]. In the present study, we wanted to focus on the effects of setting Argon plasma treatment with a power level of 80 W, a frequency of 13.56 MHz, under a 100 mTorr pressure. During 15 min within low-temperature Argon plasma at a distance of 10 mm from the porcine graft particles, we found that this treatment did not modify the biomaterial's surface (Figures 1–5); however, it improved osteoblast-like cell proliferation and differentiation (Figures 6–8). This could be explained by the removal of toxic elements (e.g., carbon) from the particles' surfaces, as EDS and XPS showed reduced carbon in PGPT. Another explanation could be the removal of micro-nano particles left during porcine particle production. Argon plasma treatment cleansed the porcine particles during their production process and led to stronger and faster interactions with cells.

In previous studies, porcine particle grafts exhibited osteoconductive properties of a bioactive bone graft material: able to provide an appropriate scaffold, allowing vascularization, promoting calcified tissue deposition, cellular infiltration, and attachment [44,45]. In the present study, PGPT produced superior results compared to the porcine control. Therefore, depending on clinical needs, PGPT can potentially be put to use in daily clinical practice [11,13]. Further studies are needed to achieve a better understanding of low-temperature plasma treatment over porcine particles for bone regeneration.

## 4. Conclusions

The present study found that the low-temperature plasma treatment of porcine graft particles did not modify the biomaterial's surface, as demonstrated by the SEM, EDS, XPS, XRD, and FTIR results. Furthermore, PGPT demonstrated higher biocompatibility by enhancing osteoblast-like cell proliferation and differentiation, according to WST-1, ALP, and real-time PCR. Further studies are needed to understand the use of PGPT for bone regeneration better.

**Author Contributions:** Conceptualization, C.S.C. and W.-J.C.; Methodology, E.S.; Software, P.Y.L.; Validation, H.-M.H., N.-C.T. and Y.-H.P.; Formal Analysis, C.S.C.; Investigation, P.Y.L.; Resources, W.-J.C.; Data Curation, E.S.; Writing—Original Draft Preparation, C.S.C.; Writing—Review and Editing, W.-J.C. and E.S.; Visualization, H.-M.H.; Supervision, N.-C.T.; Project Administration, Y.-H.P.; Funding Acquisition, Y.-H.P. and W.-J.C.

**Funding:** This research received no external funding.

**Acknowledgments:** The authors would like to thank Enago (www.enago.tw) for the English language review.

**Conflicts of Interest:** The authors declare no conflicts of interest.

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
