# Peer review of "Argon Plasma Surface Modified Porcine Bone Substitute Improved Osteoblast-Like Cell Behavior"

_coatings, doi:10.3390/coatings9020134_

Round 1

Reviewer 1 Report

Summary: The authors of this manuscript report on the plasma modification of the porcine bone particles and their physical and chemical characterization in comparison with the unmodified particles. Furthermore, an in vitro study regarding the response of MG63 osteoblasts has been performed.

The authors should consider the following points:

1. In my opinion, the manuscript title is too general and could be more suggestive if it will include the type of surface modification

2. “Abstract” section needs revision. Specifically, the phrase “Cell Viability and Alkaline Phosphatase Assays indicated an enhancement in proliferation and osteoblastic differentiation for the cells cultivated in PGPT media after 5 days, supported by Bone sialoprotein and Osteocalcin higher relative mRNA expression in the same cells” does not accurately convey the findings of the manuscript. Besides, it contains English errors. Please revise.

3. Keywords: I suggest the authors to add an additional keyword, namely “in vitro behavior”. Besides, “porcine graft” is too general and I recommend to replace it with “porcine bone graft”. The same observation is made for the whole manuscript.

4. Materials and Methods

The biology based methods (Sections 2.8-2.10) need revision.

For instance, the authors mentioned that in vitro cytotoxicity studies have been performed using the agar diffusion method in accordance with ISO 7405 (2008). However, the cell culture protocol presented seems to be a cell culture in monolayer on plastic dishes. Furthermore, they presented aspects concerning cell culture within the Section 2.8 entitled “Cell Viability Assay (MTT)”. I suggest the authors to present a distinct section devoted to the cell culture protocol for any type of assay and to remove the sentences: “The Cell Viability Assay Protocol was followed on days 1, 3, and 5, and MG-63 cells at passage 18 were cultured” and “For the assay, cells were cultured for 5 days following the same protocol previously described for the MTT assay” from the beginning of sections 2.9 and 2.10, respectively. Please revise.

Section 2.10: Do the authors measured the RNA integrity?  

5. Results

The majority of sections have to be entitled differently since, as such, they mainly suggest the methodologies used and not the results of the study.

Furthermore Section 3.6 is entitled „Cell Proliferation Assessment” but the authors present the data and the graph of the cell viability based on the MTT assay, as mentioned in the Materials and Methods section. If they want to present the data on cell proliferation it is necessary to represent in the Fig. 6 the evolution of OD values over the culture period to get information on the number of metabolically active viable cells.

I also suggest to combine sections 3.7 and 3.8 in a unique section entitled „Osteoblast differentiation”.

6. Discussion

This section looks rather as a „Results” section since the authors present again the results obtained inclusive the corresponding figures. I advise the authors to better write this section and make a more adequate reference to other works in the field pointing out the novelty and the relevance of their study.

7. Many ideas presented in the manuscript are not clearly presented and require revision. For exemple: „Cell Viability and Alkaline Phosphatase Assays indicated an enhancement in proliferation and osteoblastic differentiation for the cells cultivated in PGPT media after 5 days, supported by Bone sialoprotein and Osteocalcin higher relative mRNA expression in the same cells.” (raws 26-29); „Cell viability was assessed on days 1, 3, and 5 according to metabolic activities and was  performed according to the manufacturer’s instructions (MTT Kit, Roche Applied Science, Mannheim, Germany) (raws 138-140); „Stating that Magnesium has been known as one of the cationic substitutes for calcium in the HA lattice. Seo also reported The Ca/P ratio of the hot-pressed hydroxyapatite from pig bones was higher than that in the stoichiometric (synthetic) material.[38] (raws 336-338), etc.

Therefore, I recommend to the authors to have checked and corrected the whole manuscript by a native or proficient English speaking person.

Author Response

We would like to thank the reviewers for their careful and thorough reading of this manuscript and for the thoughtful comments and constructive suggestions, which help to improve the quality of this manuscript. We have studied these comments carefully and have made corresponding corrections that we hope will meet with your approval. The revised points are listed at the below.

We would like to express our great appreciation to the Assistant Editor, Raynor Zhou, and the reviewers for the comments on our paper. If you have any further queries, please do not hesitate to contact us.

Reviewer1:

Summary: The authors of this manuscript report on the plasma modification of the porcine bone particles and their physical and chemical characterization in comparison with the unmodified particles. Furthermore, an in vitro study regarding the response of MG63 osteoblasts has been performed.

We appreciate the positive feedback from the reviewer.

The authors should consider the following points:

1. In my opinion, the manuscript title is too general and could be more suggestive if it will include the type of surface modification

Author’s response: The manuscript title was change to “Argon Plasma Surface Modified Porcine Bone Substitute Improved Osteoblast-like Cell Behavior.”

2. “Abstract” section needs revision. Specifically, the phrase “Cell Viability and Alkaline Phosphatase Assays indicated an enhancement in proliferation and osteoblastic differentiation for the cells cultivated in PGPT media after 5 days, supported by Bone sialoprotein and Osteocalcin higher relative mRNA expression in the same cells” does not accurately convey the findings of the manuscript. Besides, it contains English errors. Please revise.

Author’s response: Line 29: (p < 0.05) was added to the sentence for more accurate finding description

3. Keywords: I suggest the authors to add an additional keyword, namely “in vitro behavior”. Besides, “porcine graft” is too general and I recommend to replace it with “porcine bone graft”. The same observation is made for the whole manuscript.

Author’s response: Line 33: keyword “in vitro behavior” was added and “porcine graft” was modified to “porcine bone graft”

4. Materials and Methods

The biology based methods (Sections 2.8-2.10) need revision.

For instance, the authors mentioned that in vitro cytotoxicity studies have been performed using the agar diffusion method in accordance with ISO 7405 (2008). However, the cell culture protocol presented seems to be a cell culture in monolayer on plastic dishes. Furthermore, they presented aspects concerning cell culture within the Section 2.8 entitled “Cell Viability Assay (MTT)”. I suggest the authors to present a distinct section devoted to the cell culture protocol for any type of assay and to remove the sentences: “The Cell Viability Assay Protocol was followed on days 1, 3, and 5, and MG-63 cells at passage 18 were cultured” and “For the assay, cells were cultured for 5 days following the same protocol previously described for the MTT assay” from the beginning of sections 2.9 and 2.10, respectively. Please revise.

Author’s response: Line 128-151: The MTT assay was only adapted from the ISO 7405 (2008), because of this, Sections 2.8-2.10 were separated and improved with more detailed protocols. Cell culture protocol was included for a clearer understanding. (Please refer to the manuscript).

Section 2.10: Do the authors measured the RNA integrity?  

Author’s response: Line 170: Total RNA was quantified by using a NanoDrop ND-1000 spectrophotometer (CapitalBio Nano Q, Beijing, China). This has been added to the manuscript.

5. Results

The majority of sections have to be entitled differently since, as such, they mainly suggest the methodologies used and not the results of the study.

Author’s response: Subsections titles were modified for more accurate description of the results.

Furthermore Section 3.6 is entitled „Cell Proliferation Assessment” but the authors present the data and the graph of the cell viability based on the MTT assay, as mentioned in the Materials and Methods section. If they want to present the data on cell proliferation it is necessary to represent in the Fig. 6 the evolution of OD values over the culture period to get information on the number of metabolically active viable cells.

Author’s response: Line 252: Figure 6 now is presented as OD values during the 5 days culture period for a representation of cell proliferation.

I also suggest to combine sections 3.7 and 3.8 in a unique section entitled „Osteoblast differentiation”.

Author’s response: Line 254:  We appreciate your suggestion and the two sections now are entitled: ‘Osteoblast differentiation’

6. Discussion

This section looks rather as a „Results” section since the authors present again the results obtained inclusive the corresponding figures. I advise the authors to better write this section and make a more adequate reference to other works in the field pointing out the novelty and the relevance of their study.

Author’s response: Section was improved and references to other works were done.

7. Many ideas presented in the manuscript are not clearly presented and require revision. For exemple: „Cell Viability and Alkaline Phosphatase Assays indicated an enhancement in proliferation and osteoblastic differentiation for the cells cultivated in PGPT media after 5 days, supported by Bone sialoprotein and Osteocalcin higher relative mRNA expression in the same cells.” (raws 26-29); „Cell viability was assessed on days 1, 3, and 5 according to metabolic activities and was  performed according to the manufacturer’s instructions (MTT Kit, Roche Applied Science, Mannheim, Germany) (raws 138-140); „Stating that Magnesium has been known as one of the cationic substitutes for calcium in the HA lattice. Seo also reported The Ca/P ratio of the hot-pressed hydroxyapatite from pig bones was higher than that in the stoichiometric (synthetic) material.[38] (raws 336-338), etc.

Author’s response: Ideas in lines 26-29 and 138-140 were improved, while 336-338 idea was removed for a better understanding of the discussion.

Therefore, I recommend to the authors to have checked and corrected the whole manuscript by a native or proficient English speaking person.

Author’s response: Manuscript was checked and corrected.

Reviewer 2 Report

The authors have performed a very extensive research on the analysis of surface modified porcine bone substitues.

1) in line 83 the authors describes the particle sizes of the samples – did they milled the pig bones after heat treatment?

2) which kind of sensor was used for the SEM imaging? There are also no SEM parameters given, which pressure, voltage was used?

3) there were no parameters given for EDS, did the authors performed line or area scans? How long was the counting period? How big was the area (if they used area scan)?

4) did the authors milled the samples for x-ray diffraction?

5) there are no parameters for the FTIR measurements given

6) the authors tested the biocompatibility after the plasma treatment, but the bone should be biocompatible (because its an bioproduct). How can you be sure that there is no immune reaction, when using the treated bone particles?

7) were the tests for biocompatibility repeated (at least 3 times)? There is no information given (only the number of samples in line 146, 153)

8) line 189, for me there is a difference between image A and C in Fig.1, did the authors measured the surface roughness (by using f.g. a 3D laserscanning microscope)?

9) how many different samples were used to get the EDS results?

10) for a better overview it should be enough to show the EDS spectra until 5 keV

11) line 224 Fig. 4 – the x-axis should be “2-theta (°)” not “angle”

12) the text in figures 2, 3, 4, 5 are to tiny, not easy to read

13) the authors wrote that they found “…significantly greater differences…” line 246: but p>0,05 is not significant

14) results in biocompatibility, are there no significant differences? Line 243, 250, 258: p>0,05;

15) discussion: line 280: “…particles showed no differences in roughness…” performed the authors surface roughness measurings? If yes, why there is no information about it in mat + meth or even in results? Did the authors measure the pore sizes?

16) why the authors did not perform mechanical testings? Especially for bone replacement its necessary to get information about the mechanical parameters like stiffness and maximum load capability

17) line 347: “…demonstrated higher biocompatibility by improving osteoblast-like cells…” why the authors didn’t repeat the experiments (in a 2nd step) with human osteoblasts (HoB) to be sure that the results can be transferred to HoB.  

Author Response

We would like to thank the reviewers for their careful and thorough reading of this manuscript and for the thoughtful comments and constructive suggestions, which help to improve the quality of this manuscript. We have studied these comments carefully and have made corresponding corrections that we hope will meet with your approval. The revised points are listed at the below.

We would like to express our great appreciation to the Assistant Editor, Raynor Zhou, and the reviewers for the comments on our paper. If you have any further queries, please do not hesitate to contact us.

Reviewer 2

The authors have performed a very extensive research on the analysis of surface modified porcine bone substitues.

The author thanks the valuable feedback from the reviewer on this manuscript.

1) in line 83 the authors describes the particle sizes of the samples – did they milled the pig bones after heat treatment?

Author’s response: Yes, the pig bones were sieved after heat treatment to a 500-1000 μm particle size (specified in line 78) and before γ-rays sterilization. Because we followed a procedure described by our laboratory in a previous study (reference 11, line 84) we decided to only briefly described the procedure and indicate the ‘milled’ process using the verb ‘filtered’ (line 83). We apologize for the confusion, ‘filtered’ verb has been changed to ‘milled’ in line 84.

2) which kind of sensor was used for the SEM imaging? There are also no SEM parameters given, which pressure, voltage was used?

Author’s response: The parameters have been updated in the manuscript with: SU3500 SEM machine.

Line 97: High resolution images were taken at a low pressure and accelerating voltage of 15.0 kV. The electron beam was focus to a fine point (primary electron) and magnified to x500 and x1500 by secondary electrons.

3) there were no parameters given for EDS, did the authors performed line or area scans? How long was the counting period? How big was the area (if they used area scan)?

Author’s response: Line 101: The parameters added to the EDS methodology: Probe measurements of samples surface topography were taken with an electron beam covering a 70 µm spectrum operating at 15 kV.

4) did the authors milled the samples for x-ray diffraction?

Author’s response: As we indicated in the sample preparation (line 84) Forty samples measuring 200 mg each were prepared for porcine graft used as the control (porcine control) and porcine graft low temperature Argon plasma treated (PGPT) used for all tests.

5) there are no parameters for the FTIR measurements given

Author’s response: Line 125: These parameters were added: All spectra were collected at room temperature at a nominal resolution of 4.00 and number of sample scans equal to 1000. The FT-IR spectra were recorded in a range of 450-4000 cm-1.

6) the authors tested the biocompatibility after the plasma treatment, but the bone should be biocompatible (because its an bioproduct). How can you be sure that there is no immune reaction, when using the treated bone particles?

Author’s response: Immune reaction is induced by the organic part of the bone. It has been previously reported that the presence of organic macromolecules disappears in the bone samples heated at 500°C. In our study the bone was heated up to 1000°C.

Murugan R, Rao KP, Kumar TS. Heat-deproteinated xenogeneic bone from slaughterhouse waste: physico-chemical properties. Bulletin of Materials Science. 2003 Aug 1;26(5):523-8.

Tadic D, Epple M. A thorough physicochemical characterisation of 14 calcium phosphate-based bone substitution materials in comparison to natural bone. Biomaterials. 2004 Mar 1;25(6):987-94.

7) were the tests for biocompatibility repeated (at least 3 times)? There is no information given (only the number of samples in line 146, 153)

Author’s response: Line 151: Yes, it was repeated 3 time.

8) line 189, for me there is a difference between image A and C in Fig.1, did the authors measured the surface roughness (by using f.g. a 3D laserscanning microscope)?

Author’s response: A 3D laser scanning microscope was not used but as we developed in the Lines 290-311 of our discussion part. The removal of micro-nano particles left on top of the porcine surface during its production it what was removed by the Argon plasma treatment without modifying the surface. This was supported by the EDS and XPS results. Due to this, is that images A and C in Fig. 1 don’t look the same

9) how many different samples were used to get the EDS results?

Author’s response: Line 105: Samples = 6. Has been added to the manuscript.

10) for a better overview it should be enough to show the EDS spectra until 5 keV

Author’s response: Thank you for the recommendation. Figure 2 has been modified for better view.

11) line 224 Fig. 4 – the x-axis should be “2-theta (°)” not “angle”

Author’s response: The x-axis was changed to “2-theta (°)”.

12) the text in figures 2, 3, 4, 5 are to tiny, not easy to read

Author’s response: Text has been improved in all the figures.

13)  the authors wrote that they found “…significantly greater differences…” line 246: but p>0,05 is not significant

Author’s response: We apologize for this misfit between the results in the writing and the P value. To keep the manuscript simply to understand the ‘(p < 0.05)’was intended to indicate statistically significant difference and be used through the manuscript. Unfortunately, during the editing was mistakenly placed the symbol in the wrong direction: ‘(p > 0.05)’. Therefore, the MTT, ALP, and Real time PCR results have this editing mistake, while all the writing indicates the real significantly greater differences from Plasma treated porcine graft, over only porcine or control groups. All the erroneous ‘(p > 0.05)’ have been corrected to the ‘(p < 0.05)’.

14) results in biocompatibility, are there no significant differences? Line 243, 250, 258: p>0,05;

Author’s response: All have significant difference, please refer to question 13 for due answer.

15) discussion: line 280: “…particles showed no differences in roughness…” performed the authors surface roughness measurings? If yes, why there is no information about it in mat + meth or even in results? Did the authors measure the pore sizes?

Author’s response: The roughness differences were established subjectively referring to the high crystalline porcine bone particle, characterized by isotropic and irregularities surface patterns. (added in the manuscript, lines 290-292)

Pores sizes were measured in the SEM micrographs. Due to the materials having the same porcine origin with no difference between them during preparation. Data is not shown in the study.

16) why the authors did not perform mechanical testings? Especially for bone replacement its necessary to get information about the mechanical parameters like stiffness and maximum load capability

Author’s response: Is our intention to do these tests in the future. But we didn’t include maximum load capability because the material is intended to be used mostly in the dental field where bone graft regeneration is done most of the time without mechanical forces over the graft material. Stiffness was not done, because we wanted to focus more in the bio-chemical part of the material. Now that we know that porcine graft treated with plasma has characteristics of a high crystalline bone graft and enhances cell’s viability and osteoblastic differentiation. We can do a next study were the mechanical testing you are proposing, can be done.

17) line 347: “…demonstrated higher biocompatibility by improving osteoblast-like cells…” why the authors didn’t repeat the experiments (in a 2nd step) with human osteoblasts (HoB) to be sure that the results can be transferred to HoB.  

Author’s response: Using human osteoblasts will be our next step in further studies, now that we know how the plasma surface treatment interacts on the porcine surface. In the present study we didn’t do it because we didn’t know how the plasma treatment was going to work on the porcine graft surface, neither how was going to affect the cells. This interaction with human osteoblasts can be also study with mechanical testing described in the question 16.

Reviewer 3 Report

Dear Authors 

Just check the English and minor changes like editing for some fresh references in your introductory and discussion heading. It will enhance your work more. 

a) Sheikh, Zeeshan, et al. "Biodegradable materials for bone repair and tissue engineering applications." Materials 8.9 (2015): 5744-5794.

b) https://www.sciencedirect.com/science/article/pii/B9780081021965000173

c) https://www.sciencedirect.com/science/article/pii/B9780081021965000185 

You can extract information from these and cite them on the text. 

Author Response

We would like to thank the reviewers for their careful and thorough reading of this manuscript and for the thoughtful comments and constructive suggestions, which help to improve the quality of this manuscript. We have studied these comments carefully and have made corresponding corrections that we hope will meet with your approval. The revised points are listed at the below.

We would like to express our great appreciation to the Assistant Editor, Raynor Zhou, and the reviewers for the comments on our paper. If you have any further queries, please do not hesitate to contact us.

Reviewer 3

Dear Authors 

Author’s response: Thank you for these comments.

Just check the English and minor changes like editing for some fresh references in your introductory and discussion heading. It will enhance your work more. 

a) Sheikh, Zeeshan, et al. "Biodegradable materials for bone repair and tissue engineering applications." Materials 8.9 (2015): 5744-5794.

Author’s response: Line 47: Reference was added in the introduction.

b) https://www.sciencedirect.com/science/article/pii/B9780081021965000173

Author’s response: Line 65: Reference was added in the introduction.

c) https://www.sciencedirect.com/science/article/pii/B9780081021965000185 

Author’s response: Line 67: Reference was added in the introduction.

You can extract information from these and cite them on the text. 

Round 2

Reviewer 1 Report

The authors partly considered my suggestions. The most critical point is that they resumed presenting their results in the Discussion section by referring to the figures shown in the Results section. To avoid this, please combine the two sections in a Results and Discussion section.

Another important point is that they failed to correct all English errors. For instance “In socket preservation with the combination of a porcine xenograft, and collagen membrane was utilized successfully maintaining the vertical and horizontal dimensions of four-wall bone defects. With sufficient bone volume for implant placement in all sites prior to implant placement [28]”, etc (Raws 276-279). Please, have the manuscript corrected by a native English speaker. 

Author Response

Dear Reviewer , 

Once again, we appreciate the positive feedback from the reviewers. We have made the corresponding corrections. Our response are in blue as below . 

Thank you for your review.

 1.The authors partly considered my suggestions. The most critical point is that they resumed presenting their results in the Discussion section by referring to the figures shown in the Results section. To avoid this, please combine the two sections in a Results and Discussion section.

Author’s response: Results and Discussion were combined in a single section.

2.Another important point is that they failed to correct all English errors. For instance “In socket preservation with the combination of a porcine xenograft, and collagen membrane was utilized successfully maintaining the vertical and horizontal dimensions of four-wall bone defects. With sufficient bone volume for implant placement in all sites prior to implant placement [28]”, etc (Raws 276-279). Please, have the manuscript corrected by a native English speaker. 

Author’s response: Manuscript was sent for English language re-editing and was done by a native English speaker (www.enago.com). The editing certificate was attached.

Reviewer 2 Report

I´m satisfied with the changes of the authors, as well as their answers to my questions. Nevertheless, I still have a few requests

Line 126; 317, 318, 319: cm-1 should be cm-1

Line 143: the authors used a WST-I Kit according to the manufacturers instructions. But they show in Line 253/Fig. 6 MTT results. In conclusion there is also the talk about MTT in Line 367. Which test was performed WST-I or MTT?

Line 151: "[...] (100x(control-sample))/control.[23]" should be (100x(control-sample))/control [23].

same in line 168: "(...) intensity.[24]" should be "(...) intensity [24]."

Line 198/Fig. 1: the scalebars should be made more visible

Line 216/Fig 2: I would show in both graphics the x-axis only up to a value of 5.0. This would increase the comparability.

I would keep a abbreviation, either FT-IR or FTIR (see Line 121. Line 309, 315, 317)

Author Response

Dear Reviewer  

Once again, we appreciate the positive feedback from the reviewers. We have made the corresponding corrections. Our response are in blue as below.  Please check them . 

Thank you for your review.

1.Line 126; 317, 318, 319: cm-1 should be cm-1

Author’s response: now Line 125; 276, 277: cm-1 were changed to cm-1.

2.Line 143: the authors used a WST-I Kit according to the manufacturers instructions. But they show in Line 253/Fig. 6 MTT results. In conclusion there is also the talk about MTT in Line 367. Which test was performed WST-I or MTT?

Author’s response: Thank you for pointing the mistakes. WST-I test was performed. The ‘MTT’ word was corrected to ‘WST-I’.

3.Line 151: "[...] (100x(control-sample))/control.[23]" should be (100x(control-sample))/control [23].

same in line 168: "(...) intensity.[24]" should be "(...) intensity [24]."

Author’s response:The manuscript was corrected . Thank you for your help! 

4.Line 198/Fig. 1: the scalebars should be made more visible

Author’s response: the scalebars were increased in size.

5.Line 216/Fig 2: I would show in both graphics the x-axis only up to a value of 5.0. This would increase the comparability.

Author’s response: Graphics now are shown with the x-axis only up to a value of 5.0.

6.I would keep a abbreviation, either FT-IR or FTIR (see Line 121. Line 309, 315, 317)

Author’s response: ‘FTIR’ abbreviation was the one kept. Thank you for your reminding.